# Low-Symmetry Phthalocyanines Bearing Carboxy-Groups: Synthesis, Spectroscopic and Quantum-Chemical Characterization

**DOI:** 10.3390/molecules27020524

**Published:** 2022-01-14

**Authors:** Dmitry A. Bunin, Nobuhle Ndebele, Alexander G. Martynov, John Mack, Yulia G. Gorbunova, Tebello Nyokong

**Affiliations:** 1A. N. Frumkin Institute of Physical Chemistry and Electrochemistry, Russian Academy of Sciences, Leninsky pr., 31, Building 4, 119071 Moscow, Russia; bunin_dm@mail.ru (D.A.B.); martynov@phyche.ac.ru (A.G.M.); 2Institute for Nanotechnology Innovation, Department of Chemistry, Rhodes University, Makhanda 6140, South Africa; g13n7164@campus.ru.ac.za (N.N.); j.mack@ru.ac.za (J.M.); 3N. S. Kurnakov Institute of General and Inorganic Chemistry, Russian Academy of Sciences, Leninsky pr., 31, 119991 Moscow, Russia

**Keywords:** phthalocyanine, UV-vis spectroscopy, MCD spectroscopy, TD-DFT

## Abstract

The synthesis and characterization of A_3_B-type phthalocyanines, **ZnPc1**–**4**, bearing bulky 2,6-diisopropylphenoxy-groups or chlorine atoms on isoindoline units “A” and either one or two carboxylic anchors on isoindoline unit “B” are reported. A comparison of molecular modelling with the conventional time dependent—density functional theory (TD-DFT) approach and its simplified sTD-DFT approximation provides further evidence that the latter method accurately reproduces the key trends in the spectral properties, providing colossal savings in computer time for quite large molecules. This demonstrates that it is a valuable tool for guiding the rational design of new phthalocyanines for practical applications.

## 1. Introduction

Phthalocyanines (Pcs) are macrocyclic ligands of particular interest because of their high stability, excellent photophysical properties and facile structural modification, which can readily be used to control the properties of corresponding materials and devices [1,2,3]. From this standpoint, low-symmetry Pcs are particularly interesting since combining various functional groups at the ligand periphery enables a rational modulation of their properties such as solubility, compatibility with nanomaterials, and nonlinear optical properties [4]. For example, low-symmetry Pcs bearing one or two carboxylic groups can act as the light-harvesting components of dye-sensitized solar cells (DSSCs) when these sensitizers are grafted on the surface of TiO_2_ or ZnO [5]. To control the aggregation and solubility of these Pcs, bulky groups can be introduced varying from relatively small *tert*-butyl groups [6] to perfluoro-*tert*-butyl substituents [7] and sterically demanding 2,6-disubstituted phenoxy-groups [8,9,10,11]. Moreover, carboxy-substituted Pcs can be conjugated with various nanomaterials, such as gold nanoparticles [12] or carbon nanotubes [13] to provide photoactive hybrid materials.

In addition to the control of aggregation, the substitution pattern should also help to align the frontier orbital energies of the sensitizer and the conduction band of metal oxides, so molecular modelling calculations can be used to guide further synthetic work in a rational manner [14] and improve device performance [15,16]. The accurate prediction of UV-visible absorption (UV-vis) spectra of low-symmetry Pc sensitizers is critical since photon absorption is the primary act in the sequence of physical processes that convert solar energy into electricity. Typically, this task is solved using the time dependent–density functional theory (TD-DFT) method [17,18,19,20], although the relatively high computational cost can limit its applicability in the context of large molecules with sterically demanding substituents. In this study, the synthesis of a series of zinc phthalocyaninates, **ZnPc1**–**4**, with bulky solubilizing diisopropylphenoxy-groups and either one or two carboxylic anchors is reported (Figure 1). Analyses of their UV-vis spectra using the classical TD-DFT approach and its simplified approximation (sTD-DFT) [21,22], demonstrate that the latter approach provides a spectacular speed-up of calculations by orders of magnitude, which is particularly useful in the theoretical treatment of large conjugated molecules such as phthalocyanines [22,23,24,25].

## 2. Results

Carboxy-substituted Pcs can be synthesized by a cross-condensation template reaction of phthalonitriles bearing bulky solubilizing groups and those functionalized either with hydroxymethyl or ester groups followed by chromatographic isolation of the low-symmetry target complexes. The CH_2_OH-substituted complexes can then be oxidized using iodoxybenzoic acid to form an aldehyde followed by oxidation to form a -COOH group by using NaClO_2_ in the presence of sulfamic acid [6,26,27]. Recently, direct oxidation of **Zn[(tBu)_3_(CH_2_OH)Pc]** to the corresponding carboxy-substituted complex was reported; the anaerobic reaction of the specified complex with KOH catalyzed by ZnO furnished the well-known **TT1** dye [28]. Ester-substituted Pcs can be hydrolyzed to form target complexes bearing -COOH groups [8,9]. This method was selected for use in this study (Figure 1).

Synthesis of A_3_B-Pcs was performed from the previously reported precursors–4,5-*bis*(2,6-diisopropylphenoxy)- or 4-chloro-5-(2,6-diisopropylphenoxy)-substituted phthalonitriles **1** and **2** for the A ring moieties, and methyl-3,4-dicyanobenzoate **3** or dimethyl-4,5-dicyanophthalate **4** on the B ring moieties. Condensation of the corresponding pairs of precursors in refluxing *n*-pentanol in the presence of Zn(OAc)_2_ and DBU afforded mixtures containing mainly the A_4_, A_3_B and A_2_B_2_ macrocyclic products. Because of transesterification, methyl groups were replaced by amyl residues, which decreased the overall polarity of the resulting complexes and hampered chromatographic separation of the A_3_B product from the A_4_ and A_2_B_2_ structures. Hydrolysis of ester bonds in **ZnPcAm1**–**4** resulted in the formation of the acid groups of the target compounds, which can be readily separated, from traces of the relatively nonpolar A_4_ and the much more polar A_2_B_2_ derivatives. Control over the separation was achieved by thin-layer chromatography (TLC) and matrix-assisted laser desorption ionization-time-of-flight (MALDI-TOF) mass spectrometry. All complexes isolated were characterized by UV-vis and ^1^H NMR spectroscopy (See Appendix A).

The UV-vis and magnetic circular dichroism (MCD) spectra of **ZnPc1**–**4** (Figure 2) were measured in DMF to suppress their aggregation properties so the spectral properties of the monomeric complexes can be compared with those of **ZnPc***, a fully symmetric A_4_ compound with sterically demanding substituents that was isolated as a byproduct during the synthesis of **ZnPc1** and **ZnPc2**. The introduction of electron-withdrawing carboxy-groups and/or chlorine atoms has only a relatively minor effect on the Q-band wavelengths, resulting in a bathochromic shift of up to 7 nm relative to that of the symmetric **ZnPc*** reference compound.

The effect of introducing the peripheral substituents was analyzed through a comparison of the experimental spectral data with the TD-DFT and sTD-DFT calculations. In contrast to previous studies of substituted phthalocyanines where bulky alkoxy and aryloxy groups were truncated and replaced with methoxy groups to save computational time [22,23,24,25], in the present work, all calculations were performed for molecules with genuine diisopropylphenoxy substituents to further demonstrate the capabilities of the sTD-DFT approach in the context of a desktop computer.

The optical properties of porphyrinoid complexes, such as phthalocyanines, can be readily conceptualized through a consideration of the molecular orbitals (MOs) associated with the 16 atom 18 π-electron inner ligand perimeter that have an M_L_ = 0, ±1, ±2, ±3, ±4, ±5, ±6, ±7, 8 sequence in ascending energy terms [29,30]. The highest occupied molecular orbital (HOMO) and lowest unoccupied molecular orbital (LUMO) have M_L_ = ±4 and ±5 angular nodal patterns, respectively, resulting in Q and B bands with ΔM_L_ = ±1 and ±9 properties at lower and higher energy. Michl [30] introduced an **a**, **s**, **-a** and **-s** nomenclature for the MOs derived from the four frontier π-MOs of a C_16_H_16_^2−^ parent perimeter (Figure 3), depending on whether there are nodal planes (**a**/**-a**) and MO coefficients (**s**/**-s**) aligned with the *y*-axis, which enables the facile comparison of the electronic structures of cyclic polyenes of differing symmetry. In the context of phthalocyanines, the peripheral benzo ring substitution introduces a second frontier π-MO with a_2u_ symmetry [29,31,32,33] that complicates the analysis of the electronic structures of phthalocyanines (Figure 4).

Spectral band deconvolution studies for a series of axially ligated Zn^II^Pc complexes by Nyokong and coworkers identified the presence of two intense overlapping Faraday **A_1_** terms in the B band region [34] that were subsequently labelled as the B_1_ and B_2_ bands [4,35,36,37,38,39,40]. These bands can be assigned primarily to the 1a_2u_→-a/-s and 1b_2u_→-a/-s one-electron transitions on the basis of the molecular modelling at the B3LYP/6-31G(d) level of theory (Table 1 and Table 2). For this reason, the 1a_2u_ MO (Figure 3) has previously been assumed to be the **s** MO of **ZnPc** in the context of Michl’s perimeter model [4,22,33,41].

The assignment of the higher energy ππ* transitions of **ZnPc1**–**4** and **ZnPc*** in the B band region is problematic due to the complicated configurational interaction that is predicted, but broadly similar trends are predicted in the TD-DFT and sTD-DFT calculations (Figure 5, Table 1 and Table 2). An extra intense band is predicted in the 300−400 nm region in the sTD-DFT calculations for **ZnPc1**–**4** and **ZnPc*** and in the TD-DFT calculations for **ZnPc1**, **ZnPc2** and **ZnPc*** (Figure 5) due to the mesomeric effects of the oxygen lone pairs of the peripheral substituents on MOs that are localized primarily on the peripheral benzo rings. A marked destabilization is predicted for these MOs, which includes the 1a_2u_ MO in the context of **ZnPc*** (Figure 4 and Figure 5, Table 1 and Table 2). It is hence reasonable to conclude that the 2a_2u_→-a/-s one-electron transitions are likely to dominate in the context of the main B band of **ZnPc*** as is predicted in the TD-DFT and sTD-DFT calculations (Figure 5, Table 1 and Table 2).

The predicted energy gaps between the MOs of **ZnPc1**–**4** that are derived from the 1a_1u_ and 2a_2u_ MOs of the **ZnPc** parent complex are significantly smaller than is the case with **ZnPc*** (Figure 4), since it is only the MOs derived from the 1b_1u_ and 1e_g_ MOs of **ZnPc** that are significantly destabilized by the presence of six or three diisopropylphenoxy groups in this context. As a result, multiple bands involving large contributions from the 1a_2u_→-a/-s and 2a_2u_→-a/-s one-electron transitions are predicted in the B band region in both the TD-DFT and sTD-DFT calculations (Table 1 and Table 2). In contrast with the complexity of the calculated spectra of **ZnPc1**–**4** and **ZnPc*** (Figure 5), only a relatively weak shoulder of absorbance is observed to the red of the main B band envelope in the experimental spectra (Figure 2). In a similar manner, only relatively minor differences are observed in the Faraday **A_1_** and pseudo-**A_1_** terms in the B band regions of the MCD spectra of **ZnPc*** and **ZnPc1**–**4** (Figure 2). These are likely to be associated with the large orbital angular momentum generated by transitions between the **s**, **-a** and **-s** MOs with M_L_ = ±4 and ±5 angular nodal properties that are largely localized on the inner ligand perimeter [30,38,39]. The MCD spectra in Figure 2 therefore provide direct spectroscopic evidence that the TD-DFT and sTDDFT calculations in Figure 5 do not provide an accurate description of the B band region. In contrast with the relatively consistent Faraday **A_1_** and pseudo-**A_1_** term band morphology that is observed in the B band region of the MCD spectra for **ZnPc*** and **ZnPc1**–**4**, substantial differences are predicted in the wavefunctions in the B band region in Table 1 and Table 2 and the simulated spectra in Figure 5. Extensive configurational interactions between a large number of ππ* excited states result in significant differences in the contributions from the 1a_2u_→-a/-s and 2a_2u_→-a/-s one-electron transitions which are highlighted in bold face in Table 1 and Table 2.

## 3. Discussion

The TD-DFT and sTD-DFT calculations (Figure 5, Table 1 and Table 2) generally reproduce trends observed in the experimental spectra in the Q band region (Figure 2 and Figure 5), which are often the most significant from the standpoint of applications such as DSSCs. The simplified approximation has the advantage of a spectacular speed-up in computation time, but the mesomeric interactions of the carboxylic acid moieties are somewhat problematic in the context of the sTD-DFT calculations. The partial replacement of the electron-donating diisopropylphenoxyl substituents of **ZnPc*** with electron-withdrawing carboxylic acid groups and chlorine atoms (Figure 1) results in a stabilization of the energies of the frontier orbitals of **ZnPc1**–**4** (Figure 4). Lower molecular symmetry results in a lifting of the degeneracy of the **-a** and **-s** MOs. Larger ΔLUMO values (Michl’s terminology for the energy splitting of the **-a** and **-s** MOs [30]) are predicted for the dicarboxylic acid substituted **ZnPc2** and **ZnPc4** complexes, since there are large MO coefficients on the peripheral carbons of the B ring moiety of the **-a** MOs, but not on those of the **-s** MOs (Figure 3). The electron-withdrawing mesomeric interaction with the carboxylic acid groups is hence expected to stabilize the **-a** MOs relative to the **-s** MOs and result in separate *x*- and *y*-polarized Q_00_ bands.

Q band splittings of 29 and 59 nm are predicted for **ZnPc2** in the TD-DFT and sTD-DFT calculations (Table 1 and Table 2), respectively, while smaller splittings of 23 and 45 nm are predicted for **ZnPc4** which has both a chlorine atoms and an isopropylphenoxyl substituent on the A_3_ benzo ring moieties. These bands are not resolved in the experimental spectra although significant broadening is observed in the Q_00_ bands of **ZnPc2** and **ZnPc4** relative to that of **ZnPc*** (Figure 2). It is hence apparent that the extent of the Q band splitting associated with the mesomeric interactions with the carboxylic acid substituents is over-estimated in the sTD-DFT calculations of **ZnPc1**–**4**. This results in a significant under-estimation of the energies of the lower energy Q_00_ bands (Table 1). In contrast, there is a systematic over-estimation of the Q_00_ band energies in the TD-DFT calculations at the B3LYP/6-31G(d) level of theory (Table 2) as has been reported previously for calculations of this type for a wide range of different porphyrins and phthalocyanine-related structures [17,33,41,44,45].

Since the envisaged application for **ZnPc1**–**4** is in DSSCs as photosensitizer dyes coating the TiO_2_ photoanode, a preliminary assessment of relevant parameters [46,47,48,49,50] was calculated on the basis of single point DFT calculations (Table 3) using the optimized geometries of the dyes at the B3LYP/SDD level of theory. Since the LUMO energies (Figure 6) are higher than that of the conduction band (CB) of TiO_2_ [46,50], injection of an electron into the TiO_2_ photoanode of the DSSC after photoexcitation should be feasible. Favorable open circuit voltage (V_oc_) values are predicted to lie in the 1.08−1.41 eV range for **ZnPc1**–**4** (Table 3). Spontaneous Gibbs free energies are predicted for the electron injection (ΔG_inj_) and dye regeneration (ΔG_regen_) processes shown in Figure 6 that are required to complete the circuit of the DSSC [46,50]. Favorable light-harvesting efficiency (LHE) values were also derived for the maxima of the Q bands using the oscillator strength (f) values from Table 1. Since these parameters appear to be promising, laboratory studies with DSSCs are already in progress to further assess the suitability of **ZnPc1**–**4** for this application.

## 4. Materials and Methods

### 4.1. Materials

The 4,5-*bis*(2,6-diisopropylphenoxy)phthalonitrile **1** [52], 4-chloro-5- (2,6-diisopropyl-phenoxy)phthalonitrile **2** [53], methyl-3,4-dicyanobenzoate **3** [54], and dimethyl-4,5-dicyanophthalate **4** [55,56] precursors were synthesized according to the previously reported procedures. Anhydrous zinc acetate was obtained by drying the corresponding dihydrate at 90 C in vacuo. 1,8-Diazabicyclo[5.4.0]undec-7-ene (DBU, Aldrich) was vacuum distilled over CaH_2_ and stored under argon. Chloroform (puris) was distilled over CaH_2_, *n*-pentanol (Aldrich) was distilled over Mg and stored under argon. Column chromatography was performed on silica (0.063–0.2 mm, Macherey-Nagel).

### 4.2. Methods

MALDI-TOF mass spectra were measured on a Bruker Daltonics Ultraflex mass spectrometer in positive ion mode with 2,5-dihydroxybenzoic acid (DHB) as a matrix. UV-visible absorption (UV-vis) spectra were recorded in CHCl_3_ on a Thermo Evolution 210 spectrometer in the 250–900 nm range. Rectangular quartz cuvettes with a 10 nm optical pathlength were used. NMR spectra were measured on a Bruker Avance-III spectrometer at a frequency of 600.13 MHz. Samples were prepared in CDCl_3_ (Cambridge Isotope Laboratories, Inc.), and filtered through a layer of alumina before use. Spectra were acquired at ambient temperatures. NMR spectra were referenced to the solvent signal (CHCl_3_, 7.26 ppm). Magnetic circular dichroism (MCD) spectra were recorded with a Chirascan plus spectrometer (Applied Photophysics, UK) equipped with a 1.0 tesla permanent magnet by using both parallel and antiparallel fields.

### 4.3. Computational Details

Geometry optimizations were carried out for unsubstituted ZnPc (**ZnPc**) as a model complex, and **ZnPc*** and **ZnPc1**–**4** at the B3LYP/6-31G(d) level of theory by using the Gaussian 09 software package [57]. Conventional TD-DFT calculations were carried out at the CAM-B3LYP/SDD level of theory since the CAM-B3LYP functional contains a long-range correction, while sTD-DFT calculations [21,22] were performed with the ORCA 5.0 package [58] using B3LYP/6-31G(d) optimized geometries. The MO energies and angular nodal patterns of **ZnPc**, **ZnPc*** and **ZnPc1**–**4** were calculated at the CAM-B3LYP/6-31G(d) level of theory [59,60,61]. The RIJCOSX approximation with auxiliary basis set def2/J was used to speed-up the sTD-DFT calculations [62,63]. Only the isomers shown for **ZnPc1**–**4** in Figure 4 are analysed in this study, since the calculated spectra of the other possible isomers were found to be broadly similar to those reported.

### 4.4. Synthesis

***Zinc 2-carboxy-9,10,16,17,23,24-hexakis(2,6-diisopropylphenoxy)phthalocyaninate*** (**ZnPc1**). A mixture of phthalonitrile **1** (300 mg, 0.6 mmol) and **3** (42 mg, 0.2 mmol), zinc acetate (76 mg, 0.4 mmol) and 1,8-diazabicyclo[5.4.0]undec-7-ene (DBU) (125 μL, 0.8 mmol) in *n*-pentanol (5 mL) was degassed and refluxed under argon overnight. Then, *n*-pentanol was evaporated, and the dark-green sticky residue was sonicated with 50 vol. % aqueous EtOH, the precipitate was filtered, and washed with 50 vol. % aqueous ethanol. The dark-green solid was washed off the filter with chloroform. After solvent evaporation, this mixture of products was separated by column chromatography on silica through a gradient elution by a mixture of chloroform with hexane (30 → 0 vol. %), then with methanol (0 → 9 vol. %). The target (pentoxycarbonyl)phthalocyanine **ZnPcAm1** was eluted with a mixture containing 10 and 0 vol. % hexane, whereas some amounts of the hydrolysed product **ZnPc1** were eluted with a mixture containing 1–5 vol. % methanol. These fractions were combined, and the solvents were evaporated to give a dark green powder, which was used for hydrolysis without further purification.

The resulting product was mixed with a saturated aqueous sodium hydroxide solution (3 mL), dry tetrahydrofuran (3 mL), and methanol (10 mL). The solution was degassed and then heated to 40 °C with vigorous stirring under argon for 1.5 h. The hydrolysis was monitored by thin-layer chromatography (TLC) [silica, hexane/acetone, 1:1 (*v/v*)]. During the hydrolysis reaction, the R*_f_* of the reaction mixture decreased to zero providing evidence for the production of the sodium salt of the target compound. The reaction mixture was then diluted with water (50 mL), acidified with concentrated hydrochloric acid to pH 3 and extracted with chloroform (3 × 20 mL). Chloroform was evaporated from organic extracts, and the dark green residue was dried in vacuo. The **ZnPc1** target complex was isolated by column chromatography on silica, eluting with dichloromethane + 0→6 vol. % methanol, followed by size-exclusion chromatography on Bio-Beads SX-1 (isocratic elution with a chloroform–methanol 2.5 vol. % mixture). Dark-green powder. Yield: 66 mg, 17%. **ZnPcAm1**: MALDI TOF, *m*/*z* calcd for C_110_H_122_N_8_O_8_Zn 1748.9, found 1748.5 [M^+^]. **ZnPc1**: MALDI TOF, *m*/*z* calcd for C_105_H_112_N_8_O_8_Zn 1678.8, found 1678.5 [M^+^]. UV-vis (DMF), *λ*_max_/nm (log *ε*) 261 (4.4), 288 (4.6), 359 (4.9), 612 (4.5), 678 (5.3). ^1^H NMR (600 MHz, Chloroform-*d* + 1/50 (*v*/*v*) pyridine-d*_5_*) δ 9.95 (s, 1H), 9.31 (d, *J* = 7.6 Hz, 1H), 8.84 (d, *J* = 6.9 Hz, 1H) (HʹPc, HʹʹPc, HʹʹʹPc), 8.40 (d, *J* = 19.2 Hz, 2H), 8.20 (d, *J* = 8.6 Hz, 2H), 8.16 (d, *J* = 4.9 Hz, 2H) (H*_Pc_*), 7.56 (d, *J* = 7.8 Hz, 6H), 7.47 (d, *J* = 7.9 Hz, 10H), 7.38 (s, 2H) (*m,p*-H*OAr*), 3.46 (q, 12H, H*_iPr_*), 1.36–1.23 (m, 72H, H*_Me_*).

***Zinc 2,3-dicarboxy-9,10,16,17,23,24-hexakis(2,6-diisopropylphenoxy)phthalocyaninate*** (**ZnPc2**). The complex was synthesized by mixing phthalonitrile **1** (300 mg, 0.6 mmol) and **4** (54 mg, 0.2 mmol), Zn(OAc)_2_ (76 mg, 0.4 mmol) and DBU (0.13 mg, 0.8 mmol) in *n*-pentanol (5 mL) using the above-described procedure for **ZnPc1**. Dark green powder. Yield: 53 mg, 14%. **ZnPcAm2**: (MALDI TOF): *m*/*z* calcd for C_116_H_132_N_8_O_10_Zn 1862.9, found 1862.8 [M^+^]. **ZnPc2**: MALDI TOF, *m*/*z* calcd for C_106_H_112_N_8_O_10_Zn 1722.8, found 1722.4 [M^+^]. UV-vis (DMF), *λ*_max_/nm (log *ε*) 289 (4.8), 360 (5.2), 615 (4.8), 682 (5.6). ^1^H NMR (300 MHz, Chloroform-*d* + 1/50 (*v*/*v*) pyridine-*d_5_*) δ 9.82 (s, 2H, Hʹ*_Pc_*), 8.37 (s, 2H), 8.19 (s, 2H), 8.14 (s, 2H) (H*_Pc_*), 7.63–7.52 (m, 6H), 7.49–7.43 (m, 12H) (*m,p*-H*_OAr_*), 3.46 (q, *J* = 6.1 Hz, 12H, H*_iPr_*), 1.28 (d, *J* = 21.7 Hz, 72H, H*_Me_*).

***Zinc 2-carboxy-9(10),16(17),23(24)-trichloro-10(9),17(16),24(23)-tris(2,6-diisopropylphenoxy) phthalocyaninate*** (**ZnPc3**). The complex was synthesized starting from phthalonitriles **2** (300 mg, 0.9 mmol) and **3** (55 mg, 0.3 mmol), Zn(OAc)_2_ (110 mg, 0.6 mmol) and DBU (0.18 mg, 1.1 mmol) in *n*-pentanol (5 mL) using the above-described procedure for **ZnPc1**. Bluish-green powder. Yield: 56 mg, 15%. **ZnPcAm3**: MALDI TOF, *m*/*z* calcd for C_74_H_71_Cl_3_N_8_O_5_Zn 1324.3, found 1322.4 [M^+^]. **ZnPc3**: MALDI TOF *m*/*z* calcd for C_69_H_61_Cl_3_N_8_O_5_Zn 1252.3, found 1254.0 [M^+^], 1276.0 [M + Na − H]^+^, 1292.0 [M + K − H]^+^. UV-vis (DMF): *λ*_max_ / nm (log *ε*): 283 (4.7), 357 (5.0), 612 (4.7), 679 (5.4). ^1^H NMR (600 MHz, Chloroform-*d* + 1/50 (*v*/*v*) pyridine-*d_5_*) δ 9.84 (s, 1H), 9.47 (m, *J* = 8.2 Hz, 1H), 9.20 (s, 1H) (Hʹ*_Pc_**,* Hʹʹ*_Pc_**,* Hʹʹʹ*_Pc_*), 8.69 (s, 2H), 8.41 (s, 2H), 8.15 (s, 2H) (H*^a^_Pc_**_,_* H*^b^_Pc_*), 7.57–7.53 (m, 3H), 7.49–7.43 (m, 6H) (*m,p*-H*_OAr_*), 3.34–3.24 (m, 6H, H*_iPr_*), 1.40–1.14 (m, 36H, H*_Me_*).

***Zinc 2,3-dicarboxy-9(10),16(17),23(24)-trichloro-10(9),17(16),24(23)-tris(2,6-diisopropyl******phenoxy)phthalocyaninate*** (**ZnPc4**). The complex was synthesized starting from phthalonitriles **2** (300 mg, 0.9 mmol) and **4** (72 mg, 0.3 mmol), Zn(OAc)_2_ (110 mg, 0.6 mmol) and DBU (0.18 mg, 1.1 mmol) in *n*-pentanol (5 mL) using the above-described procedure for **ZnPc1**. Bluish-green powder. Yield: 74 mg, 19%. **ZnPcAm4**: MALDI TOF, *m*/*z* calcd for C_80_H_81_Cl_3_N_8_O_7_Zn 1437.5, found 1437.4 [M^+^]. **ZnPc4**: MALDI TOF, *m*/*z* calcd for C_70_H_61_Cl_3_N_8_O_7_Zn 1296.3, found 1297.6 [M + H^+^]. UV-vis (DMF): *λ*_max_ / nm (log *ε*): 284 (4.6), 356 (4.9), 618 (4.6), 683 (5.1). ^1^H NMR (600 MHz, DMSO-*d*_6_) δ 9.66–9.54 (m, 2H, Hʹ*_Pc_*), 8.02 (s, 3H, H*^a^_Pc_*), 7.73 (d, *J* = 7.6 Hz, 3H), 7.63–7.56 (m, 6H) (*m,p*-H*_OAr_*), 7.31 (s, 3H, H*^b^_Pc_*), 1.32 (s, 36H, H*_Me_*).

***Zinc 2,3,9,10,16,17,23,24-octakis(2,6-di-isopropylphenoxy)phthalocyaninate*** (**ZnPc***). This product was obtained as a by-product in the synthesis of **ZnPc1** and **ZnPc2** as a green powder. MALDI TOF: *m*/*z* calcd for C_128_H_144_N_8_O_8_Zn 1987.0, found 1986.7 [M^+^]. UV-vis (DMF), *λ*_max_/nm (log *ε*): 288 (4.8), 359 (5.0), 611 (4.6), 677 (5.4). ^1^H NMR (300 MHz, Chloroform-*d +* 1/50 (*v*/*v*) CD_3_OD) δ 8.16 (s, 8H, H*_Pc_*), 7.60–7.51 (m, 8H, *m*-H*_OAr_*), 7.47 (d, *J* = 8.3 Hz, 16H, *p*-H*_OAr_*), 3.47 (q, *J* = 6.8 Hz, 16H, H*_iPr_*), 1.30 (s, 96H, H*_Me_*).

## 5. Conclusions

The rational design of novel phthalocyanine dyes for applications such as DSSCs is complicated by the challenging modelling calculations that are involved in the absence of access to a large computer cluster. This study demonstrates that the simplified sTD-DFT approach can rapidly provide useful information to predict or interpret the spectral trends observed in the Q band region of the UV-vis absorption spectra of π-extended chromophores such as **ZnPc1**–**4** and **ZnPc***. However, tt is noteworthy that the extent of the splitting of the Q band into *x*- and *y*-polarized components is over-estimated in the context of the lower symmetry **ZnPc1**–**4** complexes. It is clear from this series of test calculations for ZnPc complexes with peripheral substituents that introduce large mesomeric and inductive interactions with the π-system of the Pc ligand that the predictions made in the higher energy B band region in both TD-DFT and sTD-DFT calculations need to be treated cautiously. No significant extra insight is likely to be provided by the significantly longer calculation times associated with the conventional TD-DFT approach.

## Data Availability

Data is contained within the article or Appendix A.

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
