# Peer review of "Low-Symmetry Phthalocyanines Bearing Carboxy-Groups: Synthesis, Spectroscopic and Quantum-Chemical Characterization"

_molecules, 2022, doi:10.3390/molecules27020524_

Round 1
Reviewer 1 Report
In this manuscript, the authors report the synthesis, electronic absorption spectra, electronic structure calculations, and calculated ground-state electronic absorption of A3B type zinc phthalocyanines that bear carboxy groups. The paper highlights that using simplified TD-DFT affords similar results on predicting the electronic absorption properties of the ZnPc(s) comparing to the more expensive conventional TD-DFT methods. The employment of simplified TD-DFT may provides a convenient and “cheap” method for predicting the electronic absorptive properties of the novel metallopthalocyanines and directing future designs of molecular material for energy conversion applications.
Comments:
- The authors are encouraged to elaborate what’s the take home message from the MCD experiments from Fig 3. What’s the rational for this experiment.
- Molecular coordinates data should be provided in the SI for the geometry optimizations.
- Color-codings for compounds 1 and 3 are difficult to distinguish in Fig 3.
- The authors mention the applications of the ZnPcs for DSSC, it will be nice to have a table that summarizes the potential energy data obtained from the calculations for the ZnPcs comparing to the conduction band energies of TiO2 or ZnO.
In summary, this manuscript is recommended for publication in Molecules as a full article after minor revision.
Author Response
In this manuscript, the authors report the synthesis, electronic absorption spectra, electronic structure calculations, and calculated ground-state electronic absorption of A3B type zinc phthalocyanines that bear carboxy groups. The paper highlights that using simplified TD-DFT affords similar results on predicting the electronic absorption properties of the ZnPc(s) comparing to the more expensive conventional TD-DFT methods. The employment of simplified TD-DFT may provides a convenient and “cheap” method for predicting the electronic absorptive properties of the novel metallopthalocyanines and directing future designs of molecular material for energy conversion applications.
Comments:
- The authors are encouraged to elaborate what’s the take home message from the MCD experiments from Fig 3. What’s the rational for this experiment.
Extra text has been added to address this point. The MCD intensity mechanism results in greater intensity for the Q and B bands, since it is related to changes in orbital angular momentum on moving from the ground to the excited state. The MCD spectra of ZnPc1-4 and ZnPc* are broadly similar in the B band region. In contrast, markedly different wavefunctions for the TD-DFT and sTDDFT calculations as described in the renumbered Figure 5 and Tables 1 & 2.
- Molecular coordinates data should be provided in the SI for the geometry optimizations.
Cartesian coordinates were added to SI.
- Color-codings for compounds 1 and 3 are difficult to distinguish in Fig 3.
The figure has been redrafted to address this issue.
- The authors mention the applications of the ZnPcs for DSSC, it will be nice to have a table that summarizes the potential energy data obtained from the calculations for the ZnPcs comparing to the conduction band energies of TiO2 or ZnO.
An extra table and figure and extra text with six additional references have been added to address this point and explain why ZnPc1-4 are potentially suitable for use as photosensitizer dyes in DSSCs.
In summary, this manuscript is recommended for publication in Molecules as a full article after minor revision.
Thank you for positive evaluation of our manuscript!

Reviewer 2 Report
Gorbunova and Nyokong reported the synthesis and characterization of A3B-type phthalocyanines ZnPc1-4 bearing various functional groups. Compared with the conventional TD-DFT approach, the simplified sTD-DFT approach can accurately and rapidly provide the key trends in the spectral properties, providing colossal savings in computer time for quite large molecules. This demonstrates that it is a valuable tool for guiding the rational design of new phthalocyanines for practical applications. The whole manuscript is exceptionally well written, the organization of the manuscript is correct so the reader can easily follow the main discussion along the text. The only critical comment concerns the quality of some figures – in the final version it would be beneficial for the comfort of readers to put high-resolution images. Otherwise, the manuscript is perfect, therefore I recommend its publication after minor revision including improvement of figures quality and the addition of carbon NMR spectrums of the new compounds studied in this manuscript.
Author Response
Gorbunova and Nyokong reported the synthesis and characterization of A3B-type phthalocyanines ZnPc1-4 bearing various functional groups. Compared with the conventional TD-DFT approach, the simplified sTD-DFT approach can accurately and rapidly provide the key trends in the spectral properties, providing colossal savings in computer time for quite large molecules. This demonstrates that it is a valuable tool for guiding the rational design of new phthalocyanines for practical applications. The whole manuscript is exceptionally well written, the organization of the manuscript is correct so the reader can easily follow the main discussion along the text. The only critical comment concerns the quality of some figures – in the final version it would be beneficial for the comfort of readers to put high-resolution images. Otherwise, the manuscript is perfect, therefore I recommend its publication after minor revision including improvement of figures quality and the addition of carbon NMR spectrums of the new compounds studied in this manuscript.
Thank you for your kind words and positive evaluation of our manuscript!
The poor image quality was an artifact of converting a doc file to a pdf file. We are now resubmitting the pdf file with high quality images.
Regarding 13C NMR, unfortunately we cannot afford to prepare Pc samples concentrated enough to obtain informative well-resolved 13C NMR spectra due to aggregation of low-symmetric molecules at high concentrations.

Reviewer 3 Report
The article is devoted to the synthesis and characterization of new zinc phthalocyanines. Various spectroscopic methods (NMR, MS, UV-VIS) are used in the work, as well as the DFT calculations. The conclusions are sound and the results look interesting.
The work can be published after the elimination of some technical notes:
1) Figures 1-6 are given in low resolution. It may be an artifact of the file for review, but the quality needs to be improved in the publication. Possible it's make sense to tename Figure 2 as №Scheme №". The frame on the left side of the Figure 2 seems to be not necessary. The yellow color of the inscriptions in figure 6 is difficult to read.
2) Tables 1-2 look overwhelmed. It seems appropriate to present the necessary data in a graphical form, and publish the tables in SI.
3) It is necessary to carefully check the refs. Ref 1 starts with duplicate last names of authors:
"Lukyanets, E.A .; Nemykin, V.N .; Luk'yanets, E.A .; Nemykin, V.N. The key role of peripheral substituents in the chemistry of
phthalocyanines and their analogs. J. Porphyr. Phthalocyanines 2010, 14, 1-40, doi: 10.1142 / S1088424610001799."
Author Response
The article is devoted to the synthesis and characterization of new zinc phthalocyanines. Various spectroscopic methods (NMR, MS, UV-VIS) are used in the work, as well as the DFT calculations. The conclusions are sound and the results look interesting.
Thank you for positive evaluation of our manuscript!
The work can be published after the elimination of some technical notes:
1) Figures 1-6 are given in low resolution. It may be an artifact of the file for review, but the quality needs to be improved in the publication.
Indeed, low resolution was an artifact of file conversion, now the conversion was made with High quality presets.
Possible it's make sense to tename Figure 2 as №Scheme №". The frame on the left side of the Figure 2 seems to be not necessary.
Figure 2 has been renamed Scheme 1, the frame has been removed and all subsequent figures have also been renumbered to maintain the correct sequence.
The yellow color of the inscriptions in figure 6 is difficult to read.
The yellow inscriptions in Figure 6 (now Figure 5) were corrected to make them more readable.
2) Tables 1-2 look overwhelmed. It seems appropriate to present the necessary data in a graphical form, and publish the tables in SI.
The graphical forms of Tables 1 and 2 are provided the renumbered Figure 5 in the revised manuscript. Extra text has been added to address why the tables provide important information. We would prefer, if possible, to also have these tables in the main text to highlight the contrast between the relatively similar Faraday A1 or pseudo-A1 terms observed experimentally in the B band region of the MCD spectra of ZnPc1-4 and ZnPc*, and the extensive configurational interaction and markedly different wavefunctions that are predicted for the bands predicted in this spectral region in the TD-DFT and sTDDFT calculations.
3) It is necessary to carefully check the refs. Ref 1 starts with duplicate last names of authors:
"Lukyanets, E.A .; Nemykin, V.N .; Luk'yanets, E.A .; Nemykin, V.N. The key role of peripheral substituents in the chemistry of
phthalocyanines and their analogs. J. Porphyr. Phthalocyanines 2010, 14, 1-40, doi: 10.1142 / S1088424610001799."
The reference was corrected.